# From Molecular to Functional Effects of Different Environmental Lead Exposure Paradigms

**DOI:** 10.3390/biology11081164

**Published:** 2022-08-03

**Authors:** Liana Shvachiy, Ângela Amaro-Leal, Tiago F. Outeiro, Isabel Rocha, Vera Geraldes

**Affiliations:** 1Department of Experimental Neurodegeneration, Center for Biostructural Imaging of Neurodegeneration, University Medical Center Göttingen, 37073 Göttingen, Germany; lianashvachiy@fm.ul.pt (L.S.); tiago.outeiro@med.uni-goettingen.de (T.F.O.); 2Cardiovascular Centre of the University of Lisbon, 1649-028 Lisbon, Portugal; araqueleal@hotmail.com (Â.A.-L.); isabelrocha0@gmail.com (I.R.); 3Institute of Physiology, Faculty of Medicine of the University of Lisbon, 1649-035 Lisbon, Portugal; 4Max Planck Institute for Natural Science, 37075 Göttingen, Germany; 5Translational and Clinical Research Institute, Faculty of Medical Sciences, Newcastle University, Framlington Place, Newcastle Upon Tyne NE2 4HH, UK; 6Scientific Employee with an Honorary Contract at Deutsches Zentrum für Neurodegenerative Erkrankungen (DZNE), 37075 Göttingen, Germany

**Keywords:** environmental lead exposure, long-term memory impairment, hypertension, baroreflex impairment

## Abstract

**Simple Summary:**

Our comparative study brings new insights regarding the effects of environmental lead exposure on the cardiorespiratory and nervous systems. We show how various kinds of exposure can lead to different toxicities, with various degrees of nefarious effects. The developmental period is of utmost importance to the toxicity of environmental lead; however, we found that the duration of exposure is the prime reason for stronger effects, even though the dual effect of intermittent exposure causes greater molecular neuronal alterations.

**Abstract:**

Lead is a heavy metal whose widespread use has resulted in environmental contamination and significant health problems, particularly if the exposure occurs during developmental stages. It is a cumulative toxicant that affects multiple systems of the body, including the cardiovascular and nervous systems. Chronic lead exposure has been defined as a cause of behavioral changes, inflammation, hypertension, and autonomic dysfunction. However, different environmental lead exposure paradigms can occur, and the different effects of these have not been described in a broad comparative study. In the present study, rats of both sexes were exposed to water containing lead acetate (0.2% *w/v*), from the fetal period until adulthood. Developmental Pb-exposed (DevPb) pups were exposed to lead until 12 weeks of age (n = 13); intermittent Pb exposure (IntPb) pups drank leaded water until 12 weeks of age, tap water until 20 weeks, and leaded water for a second time from 20 to 28 weeks of age (n = 14); and the permanent (PerPb) exposure group were exposed to lead until 28 weeks of age (n = 14). A control group (without exposure, Ctrl), matched in age and sex was used. After exposure protocols, at 28 weeks of age, behavioral tests were performed for assessment of anxiety (elevated plus maze test), locomotor activity (open-field test), and memory (novel object recognition test). Metabolic parameters were evaluated for 24 h, and the acute experiment was carried out. Blood pressure (BP), electrocardiogram, and heart (HR) and respiratory (RF) rates were recorded. Baroreflex gain, chemoreflex sensitivity, and sympathovagal balance were calculated. Immunohistochemistry protocol for NeuN, Syn, Iba-1, and GFAP staining was performed. All Pb-exposed groups showed hypertension, concomitant with a decrease in baroreflex gain and chemoreceptor hypersensitivity, without significant changes in HR and RF. Long-term memory impairment associated with reactive astrogliosis and microgliosis in the dentate gyrus of the hippocampus, indicating the presence of neuroinflammation, was also observed. However, these alterations seemed to reverse after lead abstinence for a certain period (DevPb) and were enhanced when a second exposure occurred (IntPb), along with a synaptic loss. These results suggest that the duration of Pb exposure is more relevant than the timing of exposure, since the PerPb group presented more pronounced effects and a significant increase in the LF and HF bands and anxiety levels. In summary, this is the first study with the characterization and comparison of physiological, autonomic, behavioral, and molecular changes caused by different low-level environmental lead exposures, from the fetal period to adulthood, where the duration of exposure was the main factor for stronger adverse effects. These kinds of studies are of immense importance, showing the importance of the surrounding environment in health from childhood until adulthood, leading to the creation of new policies for toxicant usage control.

## 1. Introduction

Lead is one of the 10 chemicals of major public health concern [1], representing a major neurotoxin due to its vast domestic, industrial, medical, and even technological usage by the human population for more than 8000 years [2]. Even though strong public policy strategies have been implemented already, and some applications of lead have been banned—such as leaded gasoline—lead paint, for instance, is still widely used. As of 31 December 2021, only 43% of countries have confirmed that they have legally binding controls on the production, import, sale, and use of lead paints, while 40% of countries have no such laws [3]. Lead can enter the body by ingestion, inhalation, or absorption through the skin, with ingestion via food or water intake being the most common [4,5].

Two types of exposure can be described: First, and less commonly, occupational lead exposure can be defined as elevated levels of exposure to lead during a short period of time, targeting a specific group of subjects—usually at their working premises—and synonymous with acute lead exposure [5,6,7]. Second, environmental lead exposure involves a long-lasting exposure of large populations to lower levels of lead from various sources present in their living environment, which is a public health issue in non-developed countries with high levels of lead emissions and the usage of old working methods in industry and agriculture [2,8,9,10]. The chronic exposure to low levels of lead is described as a “silent epidemic”, as it causes serious cardiovascular and nervous dysfunctions, accounting for more than a million deaths per year [1,11]. Lead toxicity is characterized by multiple effects on the body, which are directly correlated with the most susceptible organs to lead accumulation in the body—namely, the heart, brain, liver, and kidneys—causing long-lasting health effects [9,12]. Lead toxicity also accounts for hematopoietic changes and effects on the reproductive system [6,13,14,15]. However, the most nefarious are cardiorespiratory changes, such as hypertension, autonomic dysfunction, and neurological effects [7,16,17,18,19,20,21,22]. Lead has the ability to cross the blood–brain barrier, first accumulating in the astrocytic cells and then moving to the other cell types in the brain, causing several molecular and functional alterations that can lead to neurodevelopmental disorders, cognitive impairment, depression, anxiety and, later, neurodegenerative disorders, such as Alzheimer’s disease [21,22,23,24,25].

Children are the most susceptible to lead poisoning, being more prone to environmental exposure as a result of their exploratory nature, and because they absorb 4−5 times more lead than adults [26,27,28,29]. The developmental period has been described as the main period for triggering lead’s nefarious health effects on the system throughout the lifetime until adulthood [30,31,32,33,34,35,36]. Despite the Centers for Disease Control and Prevention (CDC) having adopted a new upper limit of 5 µL/dL, the adverse effects on children’s health caused by lead exposure have also been described below that limit, so no safe blood lead level in children has been identified [37].

New guidelines for the prevention and management of human lead exposure are being created every year, mostly focusing on the exposure of children [1,29,38]. Despite wide scientific evidence of the long-term adverse health effects of lead, a comparative study showing the effects of different types of lead exposure—such as chronic or permanent exposure, exposure only in the development period, or intermittent exposure to lead, the latter of which has increased in recent years due to migration, exchange, and sabbatical programs—would be important to better understand which adverse effects of lead may be reversible. Additionally, few studies have been performed with a full characterization of the cardiorespiratory and neurological effects of the diverse types of lead exposure from the fetal period to adulthood. Previously, in our laboratory, we characterized some of the cardiorespiratory, autonomic function, behavioral, and molecular changes caused by intermittent lead exposure [39].

Consequently, our current, comparative study focuses on the characterization and parallel assessment of the toxicological effects of different kinds of lead exposure, namely, in animal behavior, long-term memory, metabolic parameters, basal physiological parameters, autonomic function, and molecular changes within the brain.

## 2. Materials and Methods

### 2.1. Experimental Groups

Considering that ingestion is one of the three main intake routes for bodily lead absorption, an animal model of lead exposure was developed as described previously [40,41]. Briefly, seven-day-pregnant Wistar rats (Charles River Laboratories, Chatillon-sur-Chalaronne, France) were divided into Pb-treated and control groups. In the Pb-treated group, the tap drinking water was replaced with 0.2% (p/v) lead (II) acetate solution dissolved in deionized water (Acros Organics, New Jersey, NJ, USA).

After weaning at 21 days, rat pups of both sexes were divided into 4 groups: lead solution for long-term Pb-exposed pups (developmental (DevPb): exposure to lead until 12 weeks of age, no exposure until 28 weeks (n = 13); intermittent (IntPb): exposure to lead until 12 weeks of age, no exposure (tap water) until 20 weeks, and second exposure from 20 to 28 weeks of age (n = 14); and permanent (PerPb): exposure to lead until 28 weeks of age (n = 14)), and tap water for age-matched control pups (Ctrl rats (n = 18)). All animals were subjected to the same experimental protocol to provide a comprehensive functional and morphological assessment at the endpoint of exposure. The experimental protocol was in accordance with European and national animal welfare legislation and was approved by the Ethics Committee of the Academic Medical Centre of Lisbon (CAML), Portugal.

### 2.2. Behavioral Evaluation

Two weeks before functional evaluation, animals underwent a set of standard behavioral tests to assess (i) anxiety and stress levels [42] (elevated plus maze test), (ii) spontaneous locomotor activity and exploratory behavior [43] (open-field test), and (iii) episodic long-term memory [44] (novel object recognition test). Animals were brought into the behavior testing room for at least 1 h prior to the commencement of the testing session during the experimental days. All behavioral experiments were conducted between the hours of 8 a.m. and 6 p.m. in a quiet room with dim lighting, and all animals had a four-day handling time for the researcher and testing room habituation [45]. Between animals, all behavior apparatus was cleaned with 70% ethanol. All studies were videotaped using a UV camera (Chacon, Wavre, Belgium), and the movies were subsequently analyzed using ANY-maze software (Stoelting Co., Wood Dale, IL, USA).

#### 2.2.1. Elevated plus Maze

For anxiety evaluation, we performed the elevated plus maze test [39,42,46,47]. The apparatus consists of an elevated maze with four arms (two open arms (50 × 10 cm) perpendicular to two enclosed arms of 50 × 10 × 30 cm height) that form a plus shape, elevated 50 cm from the ground. Each animal was left at the center of the maze to freely explore the maze for 5 min without prior habituation to the maze, and the percentage of time spent in open and closed arms was evaluated using the following ratio: (time spent in open or closed arms/total time) × 100 [39,46,47].

#### 2.2.2. Open-Field Exploration Test

The open-field test (OFT) provides a unique opportunity to systematically assess novel environment exploration and general locomotor activity and allows an initial indirect screening for anxiety-related behavior in rodents [45]. This apparatus consists of a square black box (measuring 67 × 67 × 57 cm in height) “virtually” divided into three concentric squares: (1) the peripheral zone (near the walls), (2) the intermediate zone, and (3) the center. We left the animals in the maze for 5 min, which was usually long enough for evaluation of the established parameters. We calculated the total travelled distance and the average velocity of the animals [32,39,43,47,48].

#### 2.2.3. Novel Object Recognition Test

With a 24 h retention interval, the novel object recognition (NOR) test was utilized to investigate long-term memory alterations in lead-exposed rats [44]. The open-field test (OFT) arena was used to conduct this test. The objects were randomized and utilized interchangeably across trials and object types, with clear and brown glass shapes proportionate to the animals’ size. In addition, their position in relation to the other objects was changed to use each object as a source of familiarity or novelty. [39,44,47,49].

The evaluation procedure was divided into three stages: habituation, training, and testing. Each animal could freely explore the open-field test (OFT) arena for 15 min in the absence of objects during the habituation period (3 consecutive days). On the fourth day of the training phase, the animal was given two to-be-familiarized items, dubbed sample objects (S and S’ objects), for 5 min. The animal was returned to its own cage for 24 h after being exposed to the sample objects. The animal was exposed to two objects for 5 min on the fifth day (the test phase): one previously encountered sample object (S), and one novel object (N) [44]. Training and testing days were recorded and analyzed by 3-point analysis (head, torso, and tail of the animal) using ANY-maze^®^ software, and only the data from the head point analysis were relevant for the exploration of the objects. Exploratory behavior was quantified as the amount of time animals spent around each object in both the training and testing phases. The numbers of approaches that included sniffing the object, rearing towards the object, or touching the object were counted. Sitting backwards to the object or crossing in front of the object without pointing the snout in the object’s direction was not considered exploration [44]. Exploration time was quantified as follows: ET (%) = (time exploring the object/overall exploring time) × 100.

The novelty index was calculated from the data obtained on the NOR testing day, as follows:(ET% Novel − ET% Sample)/(ET% Novel + ET% Sample)

This index ranges from −1 to 1, where negative values to 0 represent the absence of discrimination between the novel and familiar objects—i.e., more time exploring the sample object, or equal time exploring both objects—and 1 corresponds to the exploration of the novel object only [39,47,49].

### 2.3. Metabolic Evaluation

At 28 weeks, before the acute experiment, rats were housed for 24 h in metabolic cages to evaluate their body weight, food and liquid intake, and urine and feces production.

### 2.4. Functional Evaluation

#### 2.4.1. Acute Physiological Studies

At the end of the lead exposure protocols, at 28 weeks of age, each animal from the experimental protocol was anesthetized with sodium pentobarbital (60 mg/kg, i.p.) and maintained, when necessary and after testing the withdrawal reflex, with a 20% solution (*v/v*) of the same anesthetic. A homoeothermic blanket attached to a rectal probe kept the rectal temperature between 37.5 and 38.5 °C (Harvard Apparatus). For tracheal pressure recording and artificial ventilation, the trachea was cannulated below the larynx. Blood pressure was monitored, and saline and medication injections were injected into the femoral artery and vein, respectively. The electrocardiogram (ECG) was recorded using subcutaneous electrodes in three of the four limbs, while the heart rate was calculated using the ECG data (Neurolog, Digitimer). By retrograde cannulation of the external carotid artery, the right carotid artery bifurcation was detected, and the tip of a catheter was placed within the right carotid sinus. Lobeline (0.2 mL, 25 μg/mL, Sigma) was injected to stimulate the carotid body receptors [40]. Baroreceptors were stimulated by intravenous injection of phenylephrine (0.2 mL, 25 µg/mL, Sigma) [39,40,47]. Each provocation was separated by at least 3 min to allow for recovery to baseline values. An identical volume of saline was injected as a control at the start of the experiment and was proven to have no effect on the recorded variables.

At the beginning of the experimental protocol, and upon the stabilization of the physiological parameters, a basal recording of 10 min was taken for further autonomic evaluation. Blood pressure, ECG, heart rate, tracheal pressure, and breathing rate were all continually monitored and recorded throughout the test (PowerLab, AD Instruments).

Blood was drawn from the femoral artery at the end of the study to determine blood lead levels (BLLs) using an atomic absorption spectrophotometer (Shimadzu, Model no. AA 7000, Kyoto, Japan). After that, the animal was given an overdose of anesthesia, and the brain was removed.

#### 2.4.2. Data Acquisition and Analysis

All of the recorded variables were acquired at 1 kHz, amplified, and filtered (Neurolog, Digitimer; PowerLab, AD Instruments, Dunedin, New Zealand).

#### 2.4.3. Baro- and Chemoreceptor Reflex Analysis

To evaluate the baroreceptor reflex function, the baroreceptor reflex gain (BRG) was quantified, calculating the variation of HR in relation to mean BP variation:ΔHR⁄ΔBP
upon phenylephrine provocation. The evaluation of the chemoreceptor response elicited by the intracarotid injection of lobeline was calculated through basal respiratory frequency (RF, in cpm) before (average of 30 s) and during lobeline stimulation, or Δ chemoreflex (lob) = RFstimulation − RFbasal.

### 2.5. Immunohistochemistry (IHC)

The brains were maintained for post-fixation in 4% paraformaldehyde (PFA) in phosphate buffer (pH 7.4) solution at 4 °C overnight. After that, the brains were washed with phosphate-buffered saline (PBS) and immersed in increasing concentra tions of sucrose (15% and 30%), embedded in gelatin (7.5% gelatin in 15% sucrose solution), frozen with liquid nitrogen and 2-metilbutane (Sigma-Aldrich, Dorset, UK), and stored at −80 °C for further evaluation.

The hippocampus was identified (B = −2.92 to −5.04), and coronal slices (25 mm) were cut using a cryostat (Leica CM 3050S, Leica Microsystems, Wetzler, Germany) and collected in a 12-well plate for preservation at −20 °C in a cryoprotectant solution to assess neurodegeneration, synaptic alterations, astrogliosis, and microgliosis. The immunohistochemistry protocol was performed as described previously [39]. Briefly, slices underwent an antigen retrieval protocol [50], were permeabilized with 0.3% Triton X-100 (Sigma-Aldrich, UK), and were blocked with 5% goat serum (BioWest, France) and 1% bovine serum (VWR, USA). Tissues were immunoassayed with various primary rabbit polyclonal antibodies diluted in blocking solution overnight at 4 °C: NeuN (1:500), Syn (1:200), GFAP (1:500), and Iba-1 (1:250) (Abcam, UK). After washing with TBS, sections were incubated for 1 h at room temperature with the secondary antibody diluted in TBS—goat anti-rabbit IgG Alexa Fluor^®^ 594 (1:1000; Thermo Fisher, USA) for NeuN-stained tissues, and goat anti-rabbit IgG Alexa Fluor^®^ 488 (1:1000; Thermo Fisher, USA) for Syn-, GFAP-, and Iba-1-stained tissues—rinsed three times, and mounted on SuperFrost^®^ Microscope Slides. For nuclear staining, we used ProLong Gold Antifade with DAPI (Sigma-Aldrich, UK).

A confocal point-scanning microscope (Zeiss LSM 880 with Airyscan) was used to examine the dentate gyrus, and fluorescent images of GFAP, Iba-1, and synaptophysin were processed and quantified using Fiji [51]. GFAP- and Iba-1-stained cells were morphologically classified into distinct categories of glial cells [52,53,54,55], and positive cells for GFAP and Iba-1 were manually counted. Using the in-house program Multichannel Cell Counter RGB, the number of NeuN-positive cells (i.e., mature neurons) was computed and quantified.

### 2.6. Statistical Analysis

Unless otherwise noted, data are expressed as the mean ± SEM and shown as a composite of all subjects’ mean values. The D’Agostino–Pearson normality test was used to assess the normality distribution of continuous data, and Levene’s test was used to assess homogeneity of variance. The data comparing the four experimental groups were analyzed using one-way ANOVA with Dunnett’s multiple comparisons test. Within each group, Student’s *t*-test for paired observations was also employed to find the percentage of exploration time between objects used in the NOR test. Due to the increase in the variability with the increase in the mean, for blood lead levels, low frequencies, high frequencies, and LF/HF ratio were first converted to natural logarithms and then statistically evaluated. Data were analyzed using GraphPad Prism 9 (GraphPad Software Inc., USA). A value of *p* < 0.05 was considered statistically significant.

## 3. Results

### 3.1. Lead Exposure from the Fetal Period until Adulthood Caused Anxiety and Cognitive Impairment

#### 3.1.1. Permanent Exposure to Lead Caused Strong Anxiety without Locomotor Changes

The open-field test was performed to evaluate the locomotor and exploration activities of the animals. We observed that neither the total travelled distance (Figure 1a: Ctrl 2154 ± 197.3; DevPb 2553 ± 304.2; IntPb - 2430 ± 263.7; PerPb 2380 ± 232.5; *p* > 0.05) nor the average velocity (Figure 1b: Ctrl 11.90 ± 1.39; DevPb 15.37 ± 1.55; IntPb 14.32 ± 1.02; PerPb 12.56 ± 1.19; *p* > 0.05) of the animals was significantly affected by the presence of lead in the drinking water.

As for the anxiety behavior that was tested in the elevated plus maze test, we observed a strong decrease in the % of time spent in the open arms by the permanent exposure group of animals, as compared to controls (Figure 1c: Ctrl 14.43 ± 1.58 vs. PerPb 3.325 ± 0.84; *p* < 0.05). Correspondingly, a significant increase in the % of time spent in the closed arms was observed in the permanent exposure group (Figure 1d: Ctrl 26.15 ± 3.40 vs. PerPb 74.87 ± 1.06; *p* < 0.001). Interestingly, for the developmental and intermittent exposures, even though no significant difference was observed in the % of time spent in the open arms (Figure 1c: Ctrl 14.43 ± 1.576; DevPb 16.04 ± 3.33; IntPb 17.29 ± 2.979; *p* > 0.05), the animals showed a significant increase in the % of time spent in the closed arms (Figure 1d: Ctrl 26.15 ± 3.40 vs. DevPb 62.70 ± 4.70; *p* < 0.001; Ctrl 26.15 ± 3.40 vs. IntPb 52.23 ± 4.97; *p* < 0.001).

#### 3.1.2. Permanent and Intermittent Lead Exposures Generated Strong Long-Term Memory Impairment

The novel object recognition test is a very common test used for evaluation of memory and learning. In our study, we used this protocol for evaluation of long-term memory impairment by having a 24 h retention time between the training and testing days.

We observed that only the control group of animals recognized the novel objects as novel, as there was a significant increase in the percentage of exploration time of the novel object compared to the sample object by this group (Figure 2a: Ctrl S 39.31 ± 2.10 vs. Ctrl N 60.69 ± 2.10; *p* < 0.05), while all of the lead-exposed groups presented no time differences between the objects (Figure 2a: DevPb S 47.91 ± 3.89 vs. DevPb N 52.09 ± 3.89; IntPb S 50.25 ± 4.79 vs. IntPb N 49.75 ± 4.79; PerPb S 55.10 ± 2.86 vs. 44.90 ± 2.86, *p* > 0.05). As for the novel object recognition index that was calculated, we observed that the IntPb and PerPb groups showed a strong decrease in the values (Figure 2b: Ctrl 0.21 ± 0.04 vs. IntPb − 0.06 ± 0.08, *p* < 0.05; Ctrl 0.21 ± 0.04 vs. PerPb 0.10 ± 0.06; *p* < 0.05) and, curiously, the DevPb group did not show a significant difference when compared to the control group, even though no recognition based on the percentage of the exploration time was observed (Figure 2b: Ctrl 0.21 ± 0.04 vs. DevPb 0.04 ± 0.08, *p* > 0.05).

### 3.2. Lead Exposure from the Fetal Period until Adulthood Provoked Neuroinflammation and Synaptic Alterations without Neurodegeneration in the Dentate Gyrus Region

#### 3.2.1. All Types of Lead Exposures Caused Strong Astrocytic and Microglial Activation

The GFAP antibody is used to stain astrocytes and determine their activation. From qualitative morphological analysis with the use of relevant papers [52,53,54], we observed that all lead-exposed groups showed a significant increase in the activation of the astrocytic cells. The representative images (Figure 3a) show a higher density in the astrocytic cells, with hypertrophy of cellular processes and GFAP upregulation, which qualitatively shows the activation of these cells. These morphological changes are also complemented by the quantitative data that we obtained. We observed that all lead-exposed groups showed a significant increase in the number of GFAP-positive cells (Figure 3b: Ctrl 133.7 ± 3.180 vs. DevPb 194.3 ± 3.3, *p* < 0.001; Ctrl 133.7 ± 3.2 vs. IntPb 192.0 ± 3.1, *p* < 0.001; Ctrl 133.7 ± 3.2 vs. PerPb 195.0 ± 6.4; *p* < 0.001).

As for the Iba1 staining that marks the microglial cells (which are depicted in the representative images in Figure 3c), we observed that, morphologically, the Ctrl, DevPb, and PerPb groups showed ramified glial cells with small cell bodies and numerous long branching processes. The IntPb group, however, showed a different cell pattern, where the cells seemed to be in a reactive state without branches, and upregulation of Iba1 alluded to the reactive state of the microglia in this group of animals. Regarding the quantitative analysis, we observed that all groups also showed a significant increase in the number of Iba-1-positive cells (Figure 3d: Ctrl 16.00 ± 2.1 vs. DevPb 28.00 ± 1.2, *p* < 0.05; Ctrl 16.00 ± 2.1 vs. IntPb 42.00 ± 3.2 *p* < 0.001; Ctrl 16.00 ± 2.1 vs. PerPb 39.33 ± 3.8; *p* < 0.001).

#### 3.2.2. Intermittent Lead Exposure Caused Synaptic Loss, without Neuronal Degeneration

Synaptophysin was used as a synaptic marker. We observed from the qualitative evaluation (Figure 4a), that there seemed to be a decrease in the synaptic marker in the intermittent exposure group and an increase in the staining in the permanent exposure group. These observations were corroborated by the quantitative analysis of the fluorescence intensity of the staining. The intermittent exposure group showed a significant decrease in the fluorescence intensity when compared to the control group (Figure 4b: Ctrl 41.23 ± 4.25 vs. IntPb 18.35 ± 1.049, *p* < 0.001). Both the DevPb and PerPb groups showed no significant difference in the fluorescence intensity levels, even though a small increase was observed in both groups (Figure 4b: Ctrl 41.23 ± 4.25 vs. DevPb 51.23 ± 0.91; Ctrl 41.23 ± 4.25 vs. PerPb 57.56± 9.11, *p* > 0.05).

As for neurodegeneration that was assessed using the NeuN neuronal marker, we observed no significant differences in the morphology (Figure 4c) or the number of NeuN-positive cells, even though a small, insignificant decrease in the number was observed in the IntPb and PerPb groups (Figure 4d: Ctrl 702.0 ± 122.8 vs. DevPb 627.3 ± 80.34 vs. IntPb 540.3 ± 53.17 vs. PerPb 556.7± 58.67; *p* > 0.05).

### 3.3. Lead Exposure from the Fetal Period until Adulthood Caused an Increase in Blood Lead Levels, with no Effects on Food and Liquid Intake or Feces and Urine Production

For the evaluation of blood lead levels (Table 1), atomic absorption spectrophotometry was performed, and we observed that the IntPb and PerPb groups showed a significant increase in the levels of lead in the blood after the experimental protocol. Interestingly, the DevPb group showed a small increase, albeit a significant one, and not reaching the minimal lead levels for concern established by the WHO (5 µg/dL). As for the metabolic evaluation (Table 1) of the animals using metabolic cages for 24 h, we observed no differences in the food and liquid intake or the feces and urine production of all groups.

### 3.4. Lead Exposure from the Fetal Period until Adulthood Caused Hypertension Concomitant with Autonomic Dysfunction

#### 3.4.1. All Types of Lead Exposure Caused Hypertension without Heart Rate or Respiratory Alterations

Systolic, diastolic, and mean blood pressures were analyzed during a basal period after the acute surgery was performed. In all lead-exposed groups, we observed an increase in the systolic (Figure 5a: Ctrl 120.1 ± 10.8 vs. DevPb 156.7 ± 5.6, *p* < 0.01; Ctrl 120.1 ± 10.8 vs. IntPb 146.9 ± 6.7, *p* < 0.05; Ctrl 120.1 ± 10.8 vs. PerPb 171.1 ± 4.3, *p* < 0.001), diastolic (Figure 5a: Ctrl 90.3 ± 7.9 vs. DevPb 121.2 ± 5.8, *p* < 0.001; Ctrl 90.3 ± 7.9 vs. IntPb 110.1 ± 2.7, *p* < 0.05; Ctrl 90.3 ± 7.9 vs. PerPb 148.3 ± 4.5, *p* < 0.001) and, consequently, mean blood pressure (Figure 5a: Ctrl 104.3 ± 8.8 vs. DevPb 136.4 ± 5.2, *p* < 0.001; Ctrl 104.3 ± 8.8 vs. IntPb 124.2 ± 3.6, *p* < 0.05; Ctrl 104.3 ± 8.8 vs. PerPb 157.5 ± 4.3, *p* < 0.001). Heart rate, evaluated by electrocardiogram during the basal period, did not differ statistically (Figure 5b: Ctrl 433.5 ± 27.3 vs. DevPb 393.9 ± 11.1 vs. IntPb 404.0 ± 9.9 vs. PerPb 411.9 ± 9.7; *p* > 0.05). Respiratory frequency, evaluated from the tracheal pressure during the basal period, showed no statistical evidence of differences based on the presence of lead in all experimental groups when compared to the controls (Figure 5c: Ctrl 60.09 ± 3.6 vs. DevPb 71.87 ± 6.2 vs. IntPb 69.94 ± 2.8 vs. PerPb 71.30 ± 3.6; *p* > 0.05).

#### 3.4.2. All Lead Exposures Caused Increased Chemoreceptor Reflex Sensitivity, Permanent Lead Exposure and Intermittent Lead Exposure Caused Strong Baroreflex Impairment, and Only the Permanent Lead Exposure Triggered Sympathetic and Parasympathetic Overexcitation

Following basal evaluation of the animals during acute experiments, the animals were stimulated with phenylephrine injection in the femoral vein to evaluate their baroreceptor reflex response. We observed a significant decrease in the baroreflex gain in both the intermittent and permanent lead-exposed groups (Figure 6a: Ctrl 0.66 ± 0.07 vs. IntPb 0.38 ± 0.03, *p* < 0.001; Ctrl 0.66 ± 0.07 vs. PerPb 0.37 ± 0.03, *p* < 0.05). Interestingly, the developmental exposure to lead caused no significant change in the baroreflex gain (Figure 6a: Ctrl 0.66 ± 0.07 vs. DevPb 0.52 ± 0.04, *p* > 0.05). Regarding the chemoreceptor reflex sensitivity, all lead-exposed groups showed a significant increase in this parameter (Figure 6b: Ctrl 10.30 ± 1.48 vs. DevPb 26.85 ± 2.936, *p* < 0.01; Ctrl 10.30 ± 1.48 vs. IntPb 33.07 ± 5.03, *p* < 0.001; Ctrl 10.30 ± 1.48 vs. PerPb 24.69 ± 1.40, *p* < 0.05). Heart rate variability was assessed to evaluate the autonomic function by calculation of low and high frequencies. We observed that only the permanent lead exposure group of animals showed a significant increase in low frequencies (Figure 6c: Ctrl 1.05 ± 0.24 vs. PerPb 2.03 ± 0.32, *p* < 0.05) and high frequencies (Figure 6c: Ctrl 0.97± 0.25 vs. PerPb 7.83 ± 2.67, *p* < 0.05); however, no significant difference was observed in the LF/HF ratio (Figure 6c: Ctrl 0.88 ± 0.25 vs. PerPb 0.65 ± 0.19, *p* < 0.05). As for the developmental and intermittent lead exposure groups, we observed no significant alterations in the LF and HF bands, and no changes in the LF/HF (Figure 6c: LF: Ctrl 1.05 ± 0.24 vs. DevPb 0.81 ± 0.06, *p* > 0.05; Ctrl 1.05 ± 0.24 vs. IntPb 1.40 ± 0.24, *p* > 0.05; HF: Ctrl 0.97 ± 0.25 vs. DevPb 1.16 ± 0.17, *p* > 0.05; Ctrl 0.97 ± 0.25 vs. IntPb 1.24 ± 0.34, *p* > 0.05; LF/HF: Ctrl 0.88 ± 0.25 vs. DevPb 0.94± 0.22, *p* > 0.05; Ctrl 0.88 ± 0.25 vs. IntPb 0.47± 0.15, *p* > 0.05).

## 4. Discussion

This comparative study provides additional insight into the associations between the physiological dynamics and different profiles of lead exposure. We provide experimental evidence that lead exposure has detrimental effects on animals’ behavior, cardiorespiratory control, and astrocytic and microglial functions.

First, independently of the type of lead exposure profile, this study reveals a clear association between lead exposure, hypertension, and concomitant baroreflex gain impairment with chemoreceptor reflex hypersensitivity, similar to other conditions, such as hypertension, acute heart ischemia, or heart failure, which could link the paraventricular nucleus solitary tract pathway to lower brainstem nuclei—in particular, the PVN–NTS axis ([56,57,58,59,60,61]. The hypertension observed in this study has been well documented in earlier studies using models of lead exposure [17,62]. Moreover, since blood pressure values are the product of cardiac output (set by heart rate and stroke volume) and total peripheral vascular resistance (also set by sympathetic vasoconstrictor activity), hypertension observed in lead-exposed rats occurs through the increase in total peripheral vascular resistance. Other possible mechanisms behind hypertension are thought to be renin–angiotensin and the sympathetic nervous system [63,64], oxidative stress [65], circulating catecholamine levels [66], beta-adrenergic receptors [66], Na+/K+ ATPase [66], and endothelial factors [66], as well as renal dysfunction [67].

A higher chemoreceptor reflex sensitivity, observed in all lead-exposed groups, indicates that lead can trigger an overall alert-like reaction, which could contribute to hypertension and a tendency for a higher respiratory rate. In fact, in the permanent lead exposure group, where the duration of Pb exposure was higher, this increased sensitivity to the chemoreceptor reflex also suggests the contribution of a protective sympathoexcitatory reflex in the maintenance of oxygen homeostasis, and appears to be an important piece of the internal defense mechanisms to manage the progression of lead toxicity [68,69,70,71]. We observed a concomitant baroreflex impairment—another defense reaction [57,72,73]—suggesting that lead toxicity may impair central autonomic areas. Baroreflex impairment occurs with a chronic increase in blood pressure that causes an inverse relationship between baroreflex and resting blood pressure, and has been observed to be related to the duration of exposure and the withdrawal of lead, due to its fast resetting and rearrangement, resulting in the reversal of the impairment that occurs in the presence of lead [74,75,76].

Second, we showed an involvement of the sympathetic nervous system in the modulation of the baroreceptor reflex responses or the development of hypertension as a result of permanent exposure to lead. In fact, in this group, the overactivity of the sympathetic nervous system was concomitant with baroreceptor reflex impairment and hypertension, in accordance with previous studies that indicated sympathetic hyperactivity after chronic lead exposure [39,40,64]

Third, different profiles of lead exposure caused long-term memory impairments associated with reactive astrogliosis and microgliosis in the dentate gyrus of the hippocampus, indicating the presence of neuroinflammation. However, these alterations can be reversible after abstinence from lead for a certain period (DevPb), and are boosted when a second exposure occurs (IntPb), along with a synaptic loss. The influence of blood pressure values on several aspects of cognition has been mostly studied under conditions of hypertension. In fact, untreated subjects with essential hypertension show reduced performance in cognitive tasks, performing poorly in learning and memory tests compared to normotensive control subjects, and treatment with antihypertensive drugs improves their cognitive performance [77]. These findings suggest that behavioral deficits in lead exposure can be attributable to elevated blood pressure, and can be reversible if the hypertension is treated [77]. These neural mechanisms (i.e., central baroreceptor pathways and autonomic outflow) that can be affected by lead exposure are also involved in cognitive modulation via short-lasting changes in blood pressure and baroreceptor activation. The link between baroreceptor function and cognitive processes is bidirectional. Cognitive–attentional processes influence cardiovascular function through baroreceptor function changes that are moderated by the type of cognitive process. Conversely, a difficult arithmetic task increases heart rate and blood pressure by reducing baroreceptor reflex, leading to a rise in cerebral blood flow velocity during the task [78].

Small blood vessels are highly vulnerable to heavy metals, as they are directly exposed to the blood circulatory system, which has a comparatively higher concentration of heavy metals than other organs, resulting in associated disorders, such as dementia, cognitive disabilities, and stroke [79]. In our study, the cerebral blood flow velocities were not evaluated, but the lead exposure, independent of the type of exposure, resulted in increases in blood pressure by reducing the baroreceptor reflex. The normal heart rate found in these animals, therefore, reflects either an adaptation of the baroreflex systems to the higher pressure, or a primary dysfunction caused by lead exposure. Such diminished sensitivity might also result from decreased compliance of the arterial walls in baroreceptor regions, allowing less stretching of receptors per unit of rise in blood pressure, thus producing the damped response that we found. Nevertheless, the neural mechanisms by which baroreceptors modulate cognitive processes have not been clarified.

Lead exposure plays a key role in synaptic neurotoxicity, as shown by the permanent and intermittent exposure resulting in long-term memory impairment, with these cognitive alterations being related to its accumulation—mainly in the astroglia [80,81,82] and microglia [81]. In fact, astrocytes, along with endothelial cells, make up the blood–brain barrier, and perform homeostatic regulatory functions that are involved in the long-term potentiation that is crucial for synaptic plasticity, learning, and memory [52,54]. In this study, the morphological evaluation of astrocytes stained with GFAP in DG showed that exposure to lead enhanced the astrocytic reactivity; more specifically, a persistent maladaptive GFAP immunoreaction was enhanced within individual cells, and the density of the cells was much higher within the area, while the branching processes of the cells were hypertrophic.

Consequently, overall, long-term memory effects were more severe in the permanent exposure group and reversible in the developmental exposure group. This effect of Pb on cognitive parameters has been widely reported in chronic exposure to lead—especially in memory and learning [4,21,23,83,84]. This difference in NOR test performance could be due to lead-induced impairments of the hippocampus, in part explained by the expression of synaptophysin, astrocytes, and microglia in the dentate gyrus region. The animals intermittently exposed to lead had a decrease in the number of synapses in the hippocampus/dentate gyrus region. Moreover, permanent exposure promotes an increase in the number of synapses in the dentate gyrus. Hence, lead exposure disturbs synaptogenesis of the dentate gyrus, which could lead to impairment of synaptic plasticity in the hippocampus [20,34,85]. Interestingly, in the study presented here, blood lead concentrations in the permanent and intermittent exposure groups were much higher than those in control rats. Despite the lower blood lead levels in the developmental Pb exposure group—caused by a lead-free period of 16 weeks—the adverse cardiovascular effects were not reversible, unlike the adverse memory effects. On the other hand, permanent exposure, with higher levels of Pb in the blood, had a marked cardiovascular adverse effect, along with sympathoexcitation.

## 5. Conclusions

To conclude, we showed that the physiological impacts induced by prenatal, pre-weaning, and post-weaning Pb exposure persists until adulthood, with the permanent exposure presenting more pronounced adverse effects, which might suggest that the duration of Pb exposure is more relevant than the time of exposure. Regarding the neurobehavioral deficits, our data confirm that they were reversible, suggesting that only a continuous exposure to Pb during early life impacts cognitive performance in adulthood. This is the first study to present the characterization and comparison of physiological, autonomic, behavioral, and molecular changes caused by different degrees of environmental low-level lead exposure, from the fetal period to adulthood, where the duration of exposure was the main factor in the severity of the adverse effects observed.

## Figures and Tables

**Figure 1 biology-11-01164-f001:**
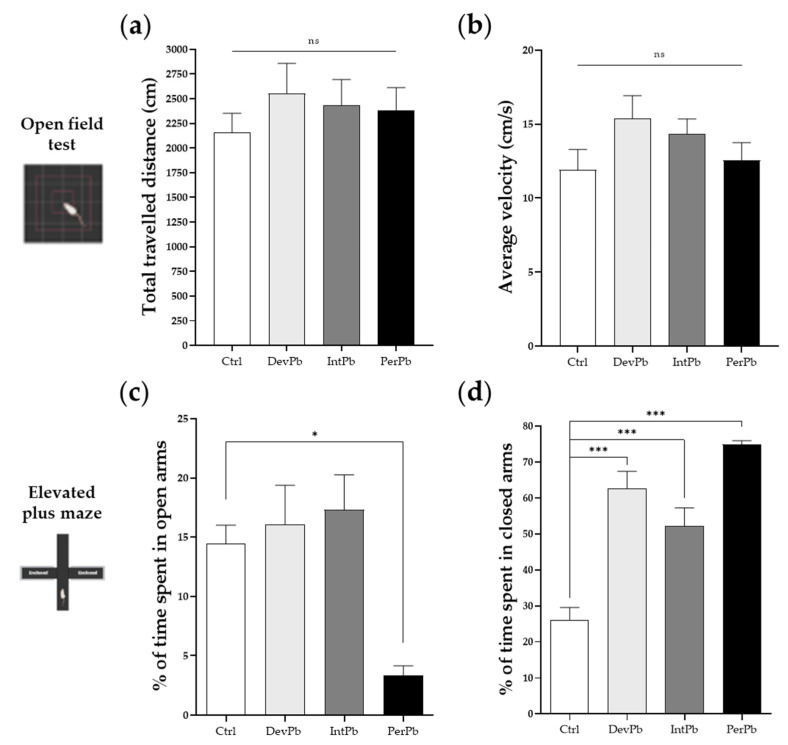
Locomotor and exploratory behaviors and anxiety assessed by the open-field test and elevated plus maze test, respectively: (**a**) Total travelled distance of the animals in the open-field test. (**b**) Average velocity in the open-field test. (**c**) Percentage of time spent in the open arms of the elevated plus maze. (**d**) Percentage of time spent by the animals in the enclosed arms of the elevated plus maze. Values are the mean ± SEM. The asterisks denote statistically significant differences between groups; ns – not significant; * *p* < 0.05; *** *p* < 0.001; one-way ANOVA, with multiple comparisons (Dunnett’s test); Ctrl n = 18; DevPb n = 13; IntPb n = 14; PerPb n = 14.

**Figure 2 biology-11-01164-f002:**
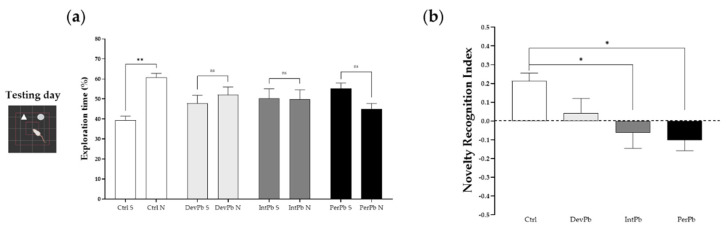
Long-term memory and learning assessment by the novel object recognition test: (**a**) Percentage of exploration time of sample (S) and novel (N) objects by each group. (**b**) Novelty recognition index calculated by the equation presented above. Values are the mean ± SEM. The asterisks denote statistically significant differences between groups; ns – not significant; * *p* < 0.05; ** *p* < 0.01; (**a**) paired Student’s t-test; (**b**) one-way ANOVA, with multiple comparisons (Dunnett’s test). Ctrl n = 18; DevPb n = 13; IntPb n = 14; PerPb n = 14.

**Figure 3 biology-11-01164-f003:**
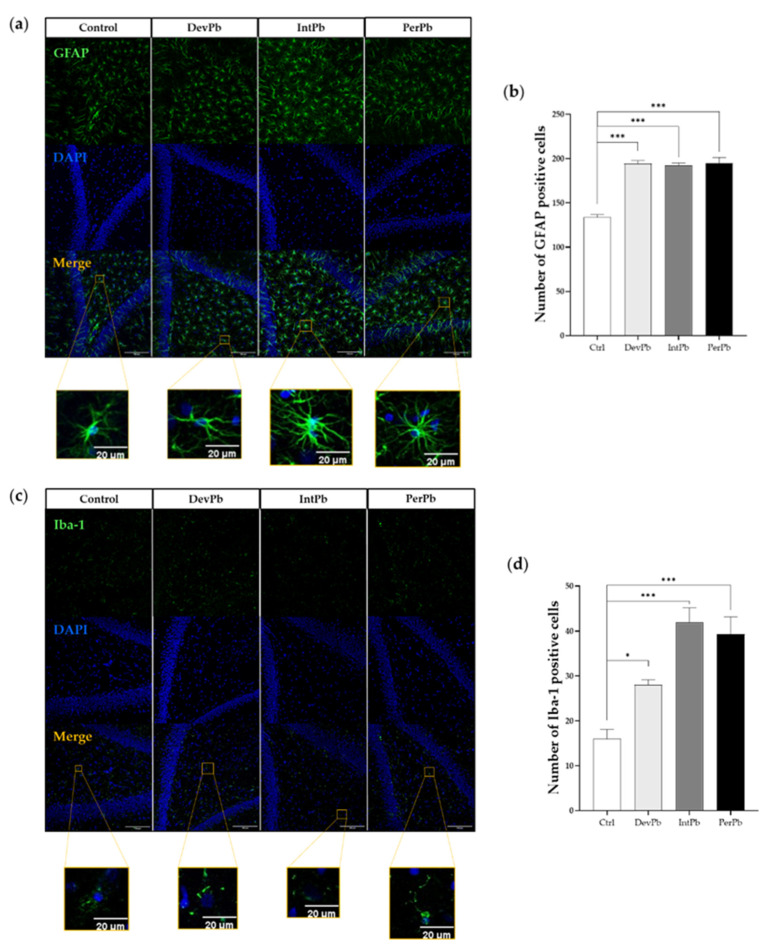
Neuroinflammation evaluation through astrocytic (GFAP) and microglial (Iba1) markers by immunohistochemistry: (**a**) Representative images of the GFAP (1:500)-stained astrocytes. (**b**) Histogram of GFAP-positive cells’ quantification. (**c**) Representative images of the Iba1 (1:250)-stained microglia. (**d**) Histogram of Iba1-positive cells quantification. Images were acquired on a confocal point scanning microscope, (Zeiss LSM 880 with Airyscan), with a 20× objective. The scale bar is 50 µm or 20 µm for stained images. Values are the mean ± SEM. The asterisks denote statistically significant differences between groups; * *p* < 0.05; *** *p* < 0.001; one-way ANOVA, with multiple comparisons (Dunnett’s test); n = 4/group.

**Figure 4 biology-11-01164-f004:**
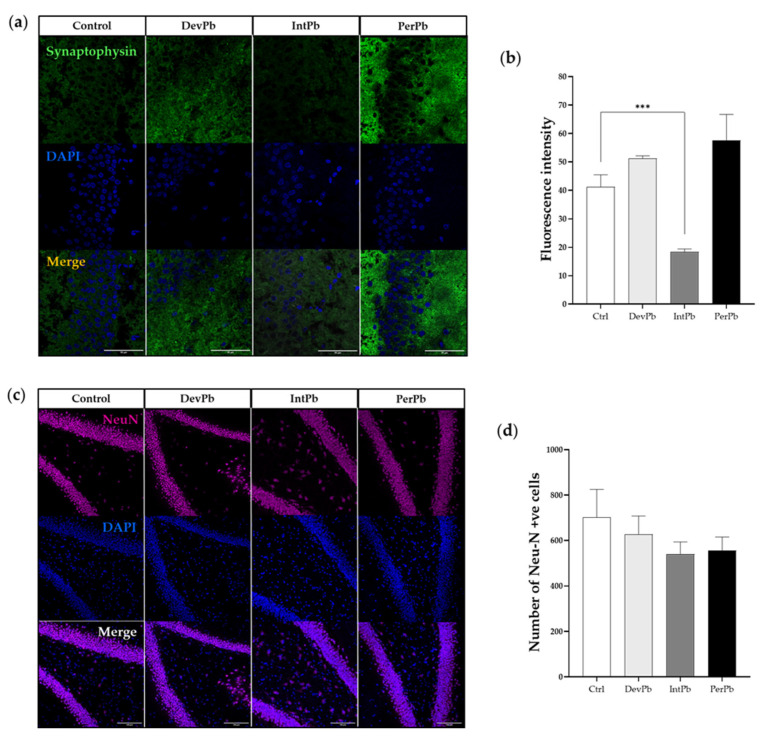
Synaptic alterations (Syn) and neurodegeneration (NeuN) results from the immunohistochemistry technique: (**a**) Representative images of the Syn (1:500)-stained pre-synapses. (**b**) Histogram of the fluorescence intensity of the Syn staining. (**c**) Representative images of the NeuN (1:500)-stained neurons. (**d**) Histogram of NeuN-positive cells’ quantification. Images were acquired using a confocal point scanning microscope (Zeiss LSM 880 with Airyscan) with 20× objective. Scale bar is 50 µm for stained images. Values are the mean ± SEM. The asterisks denote statistically significant differences between groups; *** *p* < 0.001; one-way ANOVA, with multiple comparisons (Dunnett’s test); n = 4/group.

**Figure 5 biology-11-01164-f005:**
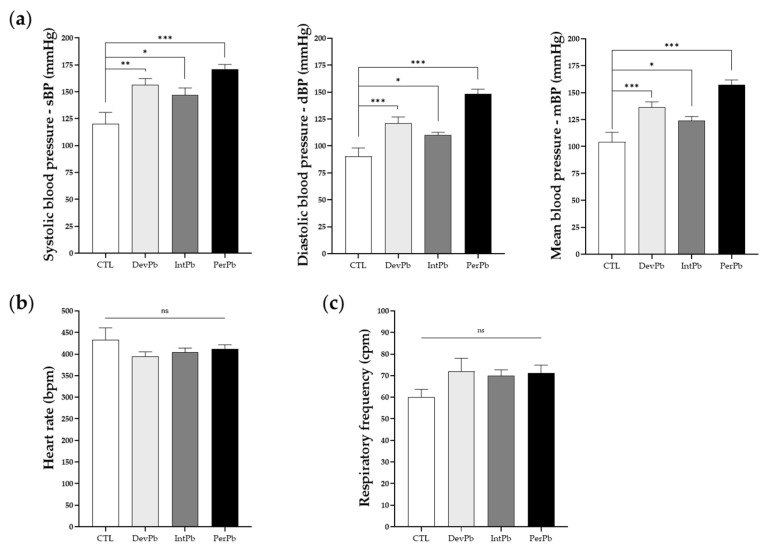
Basal cardiorespiratory data evaluated during the acute experiment before autonomic reflex stimulation: (**a**) Systolic, diastolic, and mean blood pressure assessed from the femoral artery. (**b**) Heart rate values calculated from electrocardiogram. (**c**) Respiratory rate calculated from basal tracheal pressure. Values are means ± SEM. The asterisks denote statistically significant differences between groups; ns – not significant; * *p* < 0.05; ** *p* < 0.01; *** *p* < 0.001; one-way ANOVA with multiple comparisons (Dunnett’s test). Ctrl n = 10; DevPb n = 9; IntPb n = 12; PerPb n = 12.

**Figure 6 biology-11-01164-f006:**
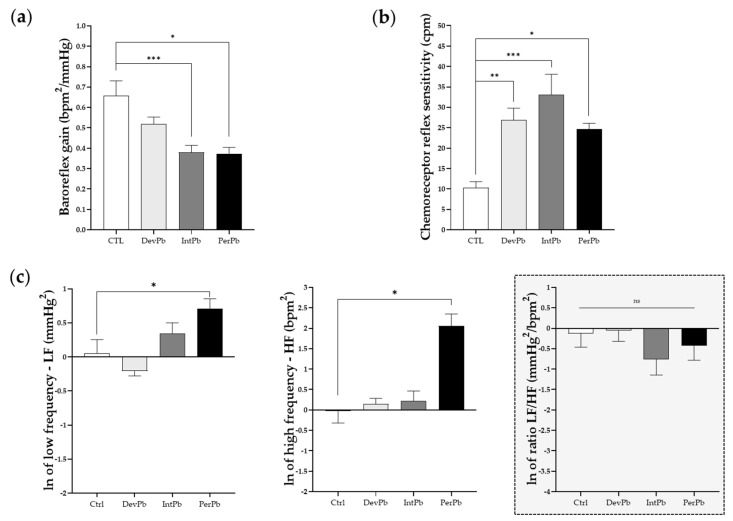
Autonomic reflexes and sympathetic and parasympathetic activity: (**a**) Baroreflex gain calculated from the variation in heart rate and blood pressure after phenylephrine stimulation (**b**) Chemoreceptor reflex sensitivity calculated from the variation in respiratory rate upon lobeline stimulation (**c**) Natural log of low frequency (LF), high frequency (HF), and ratio between LF and HF bands analyzed by heart rate variability. Values are means ± SEM. The asterisks denote statistically significant differences between groups; * *p* < 0.05; ** *p* < 0.01; *** *p* < 0.001; one-way ANOVA, with multiple comparisons (Dunnett’s test). Ctrl n = 10; DevPb n = 9; IntPb n = 12; PerPb n = 12.

**Table 1 biology-11-01164-t001:** Blood lead levels and metabolic parameters of lead-exposed groups. Values are the mean ± SEM. The asterisks denote statistically significant differences between groups; ns—not significant; one-way ANOVA, with multiple comparisons (Dunnett’s test); n = 6/group.

Group	Blood Lead Levels (μg/dL)	Weight (g)	Food Intake (g)	Liquid Intake (mL)	Produced Feces (g)	Produced Urine (mL)
Ctrl	0.6 ± 0.1	333± 41.0	22.8 ± 2.0	26.2 ± 1.8	10.8 ± 1.5	8.0 ± 0.6
DevPb	3.6 ± 0.4 ^ns^	445 ± 53.7 ^ns^	23.2 ± 2.0 ^ns^	31.2 ± 2.5 ^ns^	16.8 ± 1.0 ^ns^	8.7 ± 0.9 ^ns^
IntPb	18.4 ± 1.7 ^ns^	428 ± 50.2 ^ns^	24.0 ± 1.6 ^ns^	25.0 ± 2.2 ^ns^	10.8 ± 0.4 ^ns^	9.3 ± 2.2 ^ns^
PerPb	26.9 ± 2.2 ^ns^	434 ± 50.4 ^ns^	23.5 ± 2.8 ^ns^	29.6 ± 2.0 ^ns^	16.4 ± 2.6 ^ns^	7.5 ± 1.1 ^ns^

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
