# Peer review of "From Molecular to Functional Effects of Different Environmental Lead Exposure Paradigms"

_biology, 2022, doi:10.3390/biology11081164_

Round 1
Reviewer 1 Report
The research used rats to feed water containing lead (Pb) acetate (0.2% w/v), from fetal period until adulthood along the experimental protocol. At 28 weeks of age, behavioral tests were performed for anxiety (Elevated Plus Maze Test), locomotor activity (Open Field Test) and memory (Novel Object Recognition test) assessment. They found that Pb-exposed groups lead to hypertension, neuroinflammation, in which the duration of exposure was the main factor for stronger adverse effects.
The manuscript was well-written. I have some questions to clarify:
1. What is the concentration of 0.2% w/v water? Equal to how many ppm or mg/L?
2. The authors stated the groups of experiment rats were sex-matched, how about the numbers of female and male rats in each group? Were there significant different performance between the female and male?
3. I suggest to indicate the numbers (n=?) for each group in the result tables and figures.
Author Response
Thank you to the reviewer for your useful and constructive suggestions. We have revised the present manuscript and considered the reviewer's suggestions and comments since these changes will produce an article that better serves you and our readers.
We hope that this revised manuscript will be subject to satisfactory responses to the reviewer's comments.

Reviewer 2 Report
Overall, this has the makings of a useful paper. I hope after modification it will be accepted. I have made comments on how the paper could be brought up to publication standard.
Statistical comments
Generally the statistical analyses are sound. Dunnett’s test is appropriate as long as typical assumptions (constant variance, normality of residuals) are met. Use of a submodel where 1 df is used for looking for a trend with increasing Pb could give a more powerful test (e.g. Figure 4d). In some cases I have indicated where a transformation is required for testing (but I am happy for the diagrams to be left on the untransformed scale). The way Dunnett’s test is displayed is appropriate.
In many cases the results are clearly shown in diagrams. There are further details given in the text and often those details do not help the flow. The details would be better given in Tables – possibly as Supplementary Data
The amount of lead exposure increases across the groups – a more powerful test (If required) would use a single degree of freedom to test for that trend – that may not be necessary if the current analyses were based on the means of the pairs.
Adding an additional significance level p<0.0001 (or ****) is testing the approximation of normality, homoscedasticity. It adds little so Stay with p <0.001.
Presentation comments
Always keep a space between a number and the unit (e.g. line 208 0.2 mL)
The use of capitals is not consistent - lines 234-235 phosphate v Phosphate and 245-246 serum or Serum?
Section headings appear as a summary rather than a heading – these need attention. E.g. Line 325 – is that sentence a result summary (it should be in past tense) or a summary of prior work (needs a reference). Or is this meant to be a heading?
Detailed Comments
Title
Perhaps shorten to ‘From molecular to functional effects of different environmental lead exposure paradigms
Abstract -good.. Line 45 change ‘time of exposure’ to ‘timing of exposure’ as time of exposure could be confused with duration of exposure.’
Section 3.2.1 – section header looks like a result summary
Introduction
Line 89. Todler exposure (before 24 months) is important. This often occurs from dirty hands to mouth. A comment about that would be appropriate here.
Material and Methods
These are described in detail and match well with the results section. As the paper covers a broad range of responses, this section is of necessity longer than usual. Some areas are outside my expertise.
I assume that Dunnett’s test was applied following anova. The assumption of constant variance is fine for first 3 groups, but PerPb has a SEM double that of other treatments. Was there an outlier in that group? The test is fairly robust against changes in variance so this may not be a big issue. Knowing there is a potential violation of the assumptions I suggest changing P < 0.0001 to p < 0.001.
The tree diagrams (e.g. Figure 3 b and d) tell the story (but only use 3 ***). Lines 332 and 33 and 341 and 342 are clumsy – consider a table rows Pb group, columns data from Fig 3b and 3d. That might be supplementary information rather than in this paper. The format of Table 1 could be followed.
Section 3.2.2
Is line 352 a heading – it looks more like a summary
As mentioned above, the approximation of constant variance is questioned here. The PerPb group appears to have a larger SEM (as in previous section). Also, the moving some oof the details to a table (possibly as supplementary information).
The result in Figure 4b is counter intuitive. In other variables there has been at least an approximate correlation with amount of Pb. This requires discussion.
Analysis of data for Figure 4 was not significant overall but I suspect a single degree of freedom based on amount of Pb may be significant.
Section 3.3 Heading needs attention
Blood lead level variance increases with the mean and an anova (and hence Dunnett) is not appropriate. For that variable a logarithmic transform is required. That must be done. To show significant differences. Ok to give mean and SEM as shown.
The other variables do have approximately constant variance (as assumed in anova) – ana average SE could be used for those variables. Round weight data to the nearest g. The data on the other variables is useful, but could be summarised as the mean across the groups and much of the data be place in Supplementary data.
Section 3.4.1 Heading needs attention
First sentence strictly should be in Methods and Materials.
The data for this section are well described in Figure 5 (a and b). Figure b Rather than saying ‘heart rate was unchanged’ it would be better to say ‘ there was no statistically significant effect on heart rate’ I would like to see the details given in this section in a table – possibly in Supplementary data.
Lines 402-405 Rather than ’was unchanged’ use words like ‘did not differ statistically. In line 405 rather than ’was shown to be unaffected’ say ‘showed no statistical evidence of differences’
There was a copying problem in Figure 5a, 3rd plot. Rather than ‘physiological data’ consider ‘cardio-respiratory data’.
Section 3.4.2
Starting with Figure 6, there are faint unlabelled figures preceding a and b. They need to be labelled or omitted. The third barplot in 6c needs to be recopied.
The first 2 plots in 6c show a marked increase in variability with the mean. The underlying assumptions for anova (and hence Dunnett) are not tenable. My suggestion (requirement) is that the tests be repeated on a logarithmic scale and note added to say what was done and why. I appreciate the simplicity of showing thing on an untransformed scale – in fact the diagrams may not need to be changed if the new Dunnett’s test gives similar results. It would also be possible to do all the tests associated with Figure 6 on a log scale to give some consistency. I would suggest showing results on the linear scale – so only the number of *** would potentially change.
Line 425 ‘As for the chemoreceptor reflex sensitivity’ does not read well – The sentence should be in past tense. Similarly on line 434.
Some of the details could be given in tables in supplementary data.
Discussion
The discussion seems appropriate, but much of it is outside my area of expertise.
Author Response

(The authors gave the same response as above.)
